# Is BMI associated with anemia and hemoglobin level of women and children in Bangladesh: A study with multiple statistical approaches

**Md Kamruzzaman** [ID] *

Dept. of Applied Nutrition and Food Technology, Islamic University, Kushtia, Bangladesh

* md.kamruzzaman@anft.iu.ac.bd, mkzaman.m@gmail.com

## Abstract

### Background

The coexistence of undernutrition and obesity is an emerging problem for developing countries like Bangladesh. Anemia is another critical public health threat, prevalent predominantly among women and children. Undernutrition is linked with a higher risk of anemia, and lower dietary iron intake might be the possible reason. However, the risk of anemia among obese/overweight individuals is controversial. The study explores the relation of BMI with anemia and blood hemoglobin level among women and children in Bangladesh.

### Methods

Multiple statistical approaches were employed to nationally representative secondary data (BDHS 2011) on women (n = 5680) age 15–49 years and children (n = 2373) age 6–59 months to illuminate the relation between BMI and anemia. BMI was categorized according to the WHO recommended BMI category for Asian people. Descriptive statistics were used to measure mean hemoglobin level. Chi-square test, Pearson correlation, Two-way ANOVA, binary, ordinal, and restricted cubic splines (RCS) regression were used to study the association of BMI with anemia and hemoglobin level.

### Results

Chi-square test reveals significant association, though not intense, among BMI and anemia categories of women (15–49 years) ($\chi2 \geq 99$, p<2.2e-16 and Cramér's V = 0.0799–0.1357). From ANOVA analysis, a significant difference in blood hemoglobin level was found among women (total sample and nonpregnant) with different BMI categories (p≤0.05). Binary (Severely Underweight: OR 1.2680, 95% CI 0.755–2.161; Obese: OR 0.4038, 95% CI 0.120–1.177), Ordinal logistic regression (Severely Underweight: OR 1.337, 95% CI 0.842–2.115; Obese: OR 0.504, 95% CI 0.153–1.411) and restricted cubic spline regression (Severely Underweight: OR >1.5; Obese: OR ~0.5) reveal that the risk of anemia was higher among underweight and lower among obese/overweight women compared to normal

**Data Availability Statement:** The data are secondary and available for use by all. However, there are ethical restriction on sharing the dataset. Anyone who wishes to use the dataset can register

to do so. Details of information that are requested to access the dataset are given below. The dataset is available via the following URL: https://www.dhsprogram.com/data/available-datasets.cfm General Instructions: Exploring the above webpage, individuals first need to register as a new user to access the dataset. After registration, a new project should be created and there the country list should be provided. On the following webpages, following step-by-step, the user should be registered first, and then follow the instruction on screen after signing in to access the dataset. Contact Person and Email Address: Gbaike Ajay, Gbaike.Ajayi@icf.com For questions or comments about accessing The DHS Program data, please contact archive@dhsprogram.com.

**Funding:** This research received no specific grant from any funding agency.

**Competing interests:** There are no conflicts of interest.

women. Lower anemia risk among richest women indicates probable higher dietary iron intake among obese/overweight women.

## Conclusion

In the current study, women with overweight/obesity from Bangladesh were shown to have lower likelihood of being anemic, while underweight women more likely to be anemic. However, no relation between BMI and anemia was found among children.

## Introduction

Anemia is a condition when blood hemoglobin level or red blood cell count is below the expected cutoff points for individuals [1]. The significant consequence of anemia is lowered oxygen-carrying capacity of blood to the peripheral tissue, leading to fatigue, weakness, dizziness, and shortness of breath [2]. Globally 1.74 billion people live under the misery of anemia, and women and children are the worst sufferers [3]. Globally 58.6 (40.14–81.1) million years lived with disability in 2019 is contributed by anemia [3]. Worldwide anemia prevalence is intense in Africa and Southeast Asia, where 71% of the global mortality burden and 65% of the DALYs lost are contributed by anemia [4]. According to the indexmundi database, the prevalence of anemia in Bangladesh decreased from 55% in 1990 to 40% in 2016. However, the prevalence rate is still high [5]. Inadequate dietary iron intake, particularly more bioavailable heme iron, is endorsed as the predominant cause of anemia in Bangladesh. Moreover, poor national nutrition policy [6], household food insecurity [7], and poor dietary diversity [8] are the other causes.

In recent years, most developing countries are going through a nutrition transition, and the double burden of malnutrition is an evolving problem. The double burden of malnutrition is connected to the coexistence of undernutrition and overweight or obesity. In addition to severe undernutrition, the prevalence of obesity is increasing worldwide, and the increased prevalence of obesity is a new phenomenon, most developing countries are confronting. Obesity and overweight were previously considered a disease of only developed countries and higher socio-economic groups in developing countries, and the decreasing rate of undernutrition is not satisfactory [9–11]. In Bangladesh high prevalence of poverty and undernutrition is well documented [12]; little is available about the situation of DBM in Bangladesh. During the past two decades, Bangladesh has faced a transition socioeconomically, culturally, and demographically. The economic stability of the people of Bangladesh has increased, and urban areas have proliferated. The economic transition has triggered a shift to dietary and physical activity patterns and ultimately leads to nutrition transition and contributes to obesity and chronic diseases [13]. According to the BDHS 2017–18 report, the proportion of women who are overweight or obese has increased from 3% in 1996–97 to 32% in 2017–18 [14].

It is well evident that Higher BMI and obesity are linked with a diverse range of metabolic complications like diabetes, CVD, hypertension, and so on [15, 16]. Evidence is emerging that BMI may be linked with anemia, though the exact relation is a controversial issue and needs to be followed further. For instance, few studies reported that blood hemoglobin levels increase and the chance of becoming anemic lowered with increasing BMI [17–20]. In contrast, few studies reported the opposite trend of higher prevalence of anemia among obese/overweight women [21–27], and few reported no relation [21, 28, 29]. Studies that reported a higher risk of anemia among obese/overweight individuals stressed inflammation theory due to obesity as the vital factor of higher prevalence of anemia [30, 31]. However, the explanation is not

satisfactory, why in some studies, the higher hemoglobin level and lower risk of anemia are reported, and others reported the opposite. Besides, sometimes higher hemoglobin and the lower iron level was reported in the same study [20, 27, 32–34], which seems controversial.

Considering the increasing trend of obesity/overweight in Bangladesh and the higher prevalence and public health consequences of anemia, building up a correlation between BMI and anemia risk might help meet the challenge of obesity and anemia-related complications. Furthermore, we hypothesized that multiple statistical approaches might help elucidate those discrepancies from the previous studies. Therefore, we aim to analyze existing national representative BDHS 2011 data of Bangladesh, employing multiple statistical approaches and considering few critical factors, to study the relationship between BMI and anemia and hemoglobin level.

## Materials and methods

### Data source and study population

Bangladesh Demographic and Health Survey 6th round data (BDHS 2011) were used. Demographic and Health Survey (DHS) is conducted in more than 90 countries, and data on various population indicators are collected. The survey is a reliable source of data conducted under the sponsorships of the United States Agency for International Development (USAID) and technical assistance of ICF International of the United States. The BDHS survey was carried out from July 8 through December 27 in 2011 and was a nationwide cross-sectional survey. The survey was executed by the National Institute of Population Research and Training of the Ministry of Health and Family Welfare of Bangladesh [35]. BDHS 2011 was the first survey in Bangladesh that integrated blood hemoglobin level as the biomarker of anemia. Data on 5680 eligible women (15–49 years) and 2373 children (6–59 months) were extracted and used in the present study. Ever-married women aged between 15–49 and children aged 6–59 months were selected for blood hemoglobin level. Unmarried women were excluded, and women and children with implausible BMI (outliers) were excluded. This study was conducted according to the guidelines laid down in the Declaration of Helsinki and the ICF International approved all procedures involving research study participants, and the survey complies with the US Department of Health and Human Services regulations. The DHS survey used international ethical standards, and before interviewing all, participants were asked to a sign informed consent and/or assent. The data was collected from the DHS webpage (https://www.dhsprogram.com/data). The data is secondary and available for all to use after getting permission from the ICF International of the United States; however, restricted to sharing. The electronic approval to use the dataset for this study was received from ICF international in April 2021.

### Outcome of interest

Anemia status and hemoglobin level of women and children were selected as the dependent variable. A drop of capillary blood from the fingertip of women and fingertip or heel of children was collected and tested using Hemo Cue photometer rapid testing procedure. To adjust hemoglobin levels for altitude and smoking behavior, CDC recommended formula was used [36]. World health organization guidelines were used to classify the anemia status of women and children. For pregnant and nonpregnant women, hemoglobin level <11g/dl and <12g/dl is considered anemic, whereas for children, a level <11g/dl is considered anemic. For children and pregnant women, blood hemoglobin levels 10.0–10.9 g/dl, 7.0–9.9 g/dl, and <7g/dl are classified as Mild, Moderate, and Severe anemia. While for nonpregnant women, hemoglobin levels <11.0 g/dl, 10.0–11.9 g/dl, and 10.0–10.9 g/dl were classified as mild, moderate, and severe anemia respectively [37].

### Explanatory variables

As the aim of the study was to find out the relation between BMI and anemia status. BMI of women and BMI-for-age Z score of children were included as the principal explanatory variable. The data set contains raw BMI, and in the current study, we adopted the WHO recommended BMI category for Asian people to classify the women. The suggested category as follows: BMI <17kg/m2 as severe underweight, 17–18.49 kg/m2 as underweight, 18.5–22.99kg/m2 as normal weight, 23.0–27.49 overweight and ≥27.5 as obese [7, 38]. The raw data contain BMI Z score of children, and WHO reference standard was used to classify children. BMI SD/Z score ≥-2 to ≤2 is considered as normal weight, while <-3, <-2 to ≥-3, >2 to ≤3, and >3 is considered as severely wasted, wasted, overweight, and obese, respectively [39–41]. Other variables may confound the relation between BMI and anemia status. Therefore, few other variables like age, pregnancy status, gender, religion, and wealth index were included in the study. The wealth index is a wealth quintile developed from the household asset, representing household income and data via principal components analysis [35].

### Statistical analysis

Data were first extracted in an excel file and then transformed into a CSV file to use in R. All the statistical analyses were done in RStudio (Version 1.4.1106) based on R (Version 4.0.4) software. BMI and BMI-for-age SD score category, relative frequency, and percentages of women and children were done using descriptive statistics. A Chi-square test was performed to show the relationship between BMI category and anemia status. Chi-square ($\chi 2$), p, and Cramér's V ($\phi c$) were reported for the Chi-square test. A value of Cramér's V = 0 represents no association while 1 indicates complete association. According to the BMI category, pregnancy status, gender, and anemia category, blood hemoglobin levels of women and children were reported as mean±sd and their 95% CI. The normality of the data was tested using Q-Q-plot and Shapiro-Wilk test, and the Levene test was used to test homogeneity of variance. Two-way ANOVA was used to measure the difference in hemoglobin level among different BMI categories according to gender and pregnancy status. Tukey HSD test was used as post hoc analysis for determining the mean differences within the groups. Kruskal-Wallis rank-sum test, a nonparametric alternative of Two-way ANOVA, was used when the assumption of equal variances and normality assumption was violated. Correlation between BMI and hemoglobin level was done using the Pearson correlation coefficient test and represented on the scatter plot. Binary and ordinal logistic regression was performed to measure the association's nature, strength, and direction between anemia status and BMI and other variables. In ordinal logistic regression, anemia status was 'not anemic', 'mild', 'moderate', and 'severely anemic'. The Binary and ordinal logistic regression model was further used to predict the probability of anemia among women and children based on their BMI, pregnancy status, and gender. Additionally, restricted Cubic Splines (RCS) regression model was used to measure the odds of being anemic according to their BMI. These regression models were adjusted for age, BMI, pregnancy status, religion, and wealth index for women and age, BMI SD score, gender, and wealth index for children. A p-value of ≤0.05 was considered statistically significant.

## Results

### Chi square, ANOVA and correlation

A Chi-square test was done using crosstabulation of two and four categories of anemia with three and five categories of BMI of women and is shown in Table 1. For all crosstabulation higher, Chi-square value ($\chi 2$ ≥99) and lower p-value (2.2e-16) signify an association between

**Table 1. Prevalence of anemia among women (n = 5680) based on BMI category.**

| | Severe Underweight | Underweight | Normal Weight | Overweight | Obese | Total | χ2 |
|---|---|---|---|---|---|---|---|
| | (BMI<17) | (BMI17-18.5) | (BMI 18.5–22.99) | (BMI 23–27.49) | (BMI ≥27.50) | n(%) | (p) |
| | | | | | | | (φc) |
| **Anemia** | | | | | | | |
| *Not Anemic* | 265(8.01) | 403(12.19) | 1462(44.21) | 853(25.79) | 324(9.80) | 3307(58.22) | 108.748 |
| *Mild* | 245(12.34) | 333(16.78) | 909(45.79) | 384(19.35) | 114(5.74) | 1985(34.95) | (2.2e-16) |
| *Moderate* | 46(12.11) | 64(16.84) | 181(47.63) | 70(18.42) | 19(5.00) | 380(6.69) | 0.0799 |
| *Severe* | 2(25.00) | 2(25.00) | 3(37.50) | --- | 1((12.50) | 8(0.14) | |
| **Anemia** | | | | | | | |
| *Not anemic* | 265(8.01) | 403(12.19) | 1462(44.21) | 853(25.79) | 324(9.80) | 3307(58.22) | 104.59 |
| *Anemic* | 293(12.35) | 399(16.81) | 1093(46.06) | 454(19.13) | 134(5.65) | 2373(41.78) | (2.2e-16) 0.1357 |
| **Total [n(%)]** | 558(9.82) | 802(14.12) | 2555(44.98) | 1307(23.01) | 458(8.06) | 5680(100) | |

| | Underweight | | Normal Weight | | Overweight/Obese | | Total | χ2 |
|---|---|---|---|---|---|---|---|---|
| | (BMI<18.5) | | (BMI 18.5–22.99) | | (BMI ≥ 23) | | n (%) | (p) |
| | n (%) | | n (%) | | n (%) | | | (φc) |
| **Anemia** | | | | | | | | |
| *Not Anemic* | 668(20.20) | | 1462(44.21) | | 1177(35.59) | | 3307(58.22) | 101.879 |
| *Mild* | 578(29.12) | | 909(45.79) | | 498(25.09) | | 1985(34.95) | (2.2e-16) 0.0947 |
| *Moderate* | 110(28.95) | | 181(47.63) | | 89(23.42) | | 380(6.69) | |
| *Severe* | 4(50.00) | | 3(37.50) | | 1(12.50) | | 8(0.14) | |
| **Anemia** | | | | | | | | |
| *Not anemic* | 668(20.20) | | 1462(44.21) | | 1177(35.59) | | 3307(58.22) | 99.3745 |
| *Anemic* | 692(29.16) | | 1093(46.06) | | 588(24.78) | | 2373(41.78) | (2.2e-16) 0.1323 |
| **Total** | 1360(23.94) | | 2555(44.98) | | 1765(31.07) | | 5680(100) | |

BMI and anemia category. However, the lower Cramér's V (0.0799–0.1357) indicates the association is not complete or robust. The findings of the Chi-square test for children are shown in Table 2. Similar crosstabulation for children was done as was for women. The Chi-square test result for children was insignificant with a p-value >0.05, and Cramér's V near zero indicates an insignificant association.

According to their pregnancy status and BMI categories, the mean hemoglobin level of all women (n = 5680) is presented in Table 3 and Fig 1. Two-way ANOVA analysis shows that hemoglobin level was significantly different among different BMI categories (p≤0.05). The blood hemoglobin level of severely underweight women was 11.787±1.42 g/dl, significantly lowered from obese women with blood hemoglobin level 12.487±1.38g/dl. However, among pregnant women, the difference was not significant (p>0.05). The mean hemoglobin level for all children (n = 2373) and BMI category, gender, and anemia status are shown in Table 4 and Fig 2. The mean hemoglobin level of severely wasted children (10.535±1.273 gm/dl) was lower than obese children (11.170±1.584), though not significant (p>0.05). The hemoglobin levels of male and female children were not significantly different (p>0.05). The correlation between hemoglobin level and BMI for women is shown in Fig 3. The Pearson correlation coefficient was lower (0.15), indicating a weak association between hemoglobin level and BMI; however, the association was significant (p≤0.05). The Pearson correlation coefficient, when blood hemoglobin level and BMI Sd score were plotted for children, was zero (R = 0.027), indicating an insignificant correlation (Fig 4). The 95% CI of hemoglobin level of women according to their BMI category, pregnancy status, and anemia category are graphically presented in Fig 5. The 95% CI were overlapped with other groups, representing the insignificant difference. It is clear from Fig 5 that for all women (n = 5680) and nonpregnant women (n = 5323), the 95% CI of severely

**Table 2. Prevalence of anemia among children (n = 2373) based on BMI SD score category.**

| | Severely Wasted | Wasted | Normal Weight | Overweight | Obese | Total | χ2 |
|---|---|---|---|---|---|---|---|
| | (BMIZ<-3) | (BMIZ<-2to≥-3) | (BMIZ ≥ -2to≤ 2) | (BMIZ >2to≤ 3) | (BMIZ>3) | n(%) | (p) |
| | n(%) | n(%) | n(%) | n(%) | n(%) | | (φc) |
| **Anemia** | | | | | | | |
| *Not Anemic* | 26(1.10) | 120(5.06) | 986(41.55) | 13(0.55) | 12(0.51) | 1157(48.76) | 12.5725 |
| *Mild* | 19(0.80) | 67(2.82) | 595(25.07) | 6(0.25) | 2(0.08) | 689(29.03) | (0.4009) |
| *Moderate* | 22(0.93) | 53(2.23) | 427(17.99) | 5(0.21) | 3(0.13)) | 510(21.49) | (0.0420) |
| *Severe* | 0(0) | 00(00) | 17(0.72) | 00(0.0) | 0(0.00) | 17(0.72) | |
| **Anemia** | | | | | | | |
| *Not anemic* | 26.00(1.10) | 120(5.06) | 986(41.55) | 13(0.55) | 12(0.51) | 1157(48.76) | 6.3314 |
| *Anemic* | 41.00(1.73) | 120(5.06) | 1039(43.78) | 11(0.46) | 5(0.21) | 1216(51.24) | (0.1757) |
| | | | | | | | (0.517) |
| **Total** | 67(2.82) | 240(10.11) | 2025(85.34) | 24(1.01) | 17(0.72) | 2373(100) | |
| | **Wasted** | | **Normal Weight** | **Overweight/Obese** | | **Total** | χ2 |
| | (BMIZ<-2) | | (BMIZ≥ -2to≤ 2) | (BMIZ >2)) | | *n (%)* | **(p)** |
| | n (%) | | n (%) | n (%) | | | (φc) |
| **Anemia** | | | | | | | |
| *Not Anemic* | 146(47.56) | | 986(48.69) | 25(60.98) | | 1157(48.76) | 7.2074 |
| *Mild* | 86(28.01) | | 595(29.38) | 8(19.51) | | 689(29.03) | (0.3021) |
| *Moderate* | 75(24.43) | | 427(21.09) | 8(19.51) | | 510(21.49) | (0.0390) |
| *Severe* | 0(0) | | 17(0.84) | 00(00) | | 17(0.72) | |
| **Anemia** | | | | | | | |
| *Not anemic* | 146(47.56) | | 986(48.69) | 25(60.98) | | 1157(48.76) | 2.6304 |
| *Anemic* | 161(52.44) | | 1039(51.31) | 16(39.02) | | 1216(51.24) | (0.2684) |
| | | | | | | | (0.0333) |
| **Total** | 307(12.94) | | 2025(85.3) | 41(1.73) | | 2373(100) | |

underweight women was lowered than obese women without overlapping. The 95% CI of hemoglobin level of children according to different BMI SD score categories is shown in Fig 6.

## Regression model

Binary and ordinal logistic regression analysis for women is shown in Table 5. The binary logistic regression shows that the odds of being anemic were higher among severely underweight (OR1.401, 95%CI 1.163–1.589) and underweight (OR1.279, 95%CI 1.088–1.502) women compared to women with normal weight. On the other hand, the odds of suffering from anemia were lowered among overweight (OR 0.732, 95% CI 0.638–0.794) and obese (OR 0.562, 95% CI 0.393–0.794) women compared to women with average weight. The binary logistic regression model was adjusted for age, pregnancy status, religion, and wealth index, and both the adjusted and crude OR shown a similar result (Table 5). Pregnant women (OR 1.367, 95% CI 1.102–1.694) were likely to have higher odds of being anemic than nonpregnant women. In addition, women with the poorest wealth quintile (OR 1.281, 95% CI 1.078–1.523) were more likely to be anemic compared to women of the middle wealth quintile than women with the richest wealth quintile (OR 0.626, 95% CI 0.529–0.739). As women, the chance of being anemic was more among severely wasted (OR 1.268, 95% CI 0.755–2.161) than obese children (OR 0.403, 95% CI 0.120–1.177), when compared with normal children, though not significant.

The ordinal logistic regression shows that for severely underweight women, the odds of severe anemia (i.e., mild or moderate anemia versus not anemic) was 1.371 (95% CI

**Table 3. Blood hemoglobin level of women (n = 5680) based on BMI category and pregnancy status.**

| | Blood Hemoglobin Concentration (gm/dl) | | |
|---|---|---|---|
| | **Pregnant[#]** | **Non-Pregnant[#]** | **Total** |
| | [mean ± sd (95%CI)] | [mean ± sd (95%CI)] | [mean ± sd (95%CI)] |
| *BMI Classification* | | | |
| *Severe Underweight (BMI<17)* | 10.705±1.376(9.984–11.427)[a] | 11.823±1.409(11.698–11.949)[a] | 11.787±1.42 (11.669–11.905)[a] |
| *Underweight (BMI17-18.5)* | 10.615±1.354(10.152–11.078)[a] | 11.910±1.324(11.811–12.010)[a] | 11.847±1.353 (11.754–11.941)[a] |
| *Normal Weight (BMI 18.5–22.99)* | 11.005±1.2277(10.806–11.203)[a] | 12.139±1.314(12.083–12.194)[b] | 12.059±1.343(12.007–12.111)[b] |
| *Overweight (BMI 23–27.49)* | 11.048±1.318(10.757–11.340)[a] | 12.404±1.349(12.342–12.484)[c] | 12.311±1.390(12.235–12.386)[c] |
| *Obese (BMI ≥27.50)* | 11.487±1.097(11.062–12.911)[a] | 12.559±1.375(12.422–12.697)[c] | 12.487±1.383(12.361–12.614)[c] |
| **Anemia Classification** | | | |
| *Not anemic* | 12.021±0.760(11.911–12.132) | 13.055±0.793(13.028–13.083) | 12.998±0.825(12.970–13.027) |
| *Anemic* | 09.937±0.781(09.821–10.053) | 10.907±0.920(10.869–10.946) | 10.836±0.956(10.798–10.874) |
| **Anemia** | | | |
| *Not Anemic* | 12.021±0.760(11.911–12.132) | 13.055±0.793(13.028–13.083) | 12.998±0.825(12.970–13.027) |
| *Mild* | 10.510±0.284(10.453–10.568) | 11.200±0.519(11.177–11.224) | 11.168±0.531(11.144–11.191) |
| *Moderate* | 09.271±0.632(09.135–9.410) | 9.172±0.651(9.099–9.246) | 9.194±0.647(9.129–9.259) |
| *Severe* | --- | 6.425±0.539(6.051–6.798) | 6.425±0.539(6.051–6.798) |
| **Total** | 11.000±1.296(10.865–11.134) | 12.168±1.355(12.132–12.205) | 12.095±1.381(12.059–12.131) |

N.B.: # indicate hemoglobin level between pregnant and non-pregnant women is significantly different (p≤0.0001), Each column with different superscript indicates significant difference (p≥0.05) in hemoglobin level.

1.147–1.639) times higher than that of women who were with average weight. In contrast, for women who were obese, the odds of being severe anemia (i.e., mild or moderate anemia versus not anemic) to apply is 0.547 (95% CI 0.383–0.771) times that of women with normal weight, holding constant all other variables. The result from the wealth index shows that women from the poorest wealth index (OR 1.152, 95% CI 0.974–1.363) were more likely to be severe anemic (i.e., mild or moderate anemia versus not anemic) than women of middle wealth quintile, whereas women from the wealthiest wealth quintile were less likely (OR 0.766, 95% CI 0.640–0.902). The ordinal regression model reported that the odds of being anemic are higher among severely wasted children (OR 1.337, 95% CI 0.842–2.115) than obese children (OR 0.504, 95% CI 0.153–0.1.411) when compared to children with average weight. Women from religion Hinduism (OR 1.426, 95%CI 1.210–1.677) were more prone to suffer from anemia than women from religion Islam.

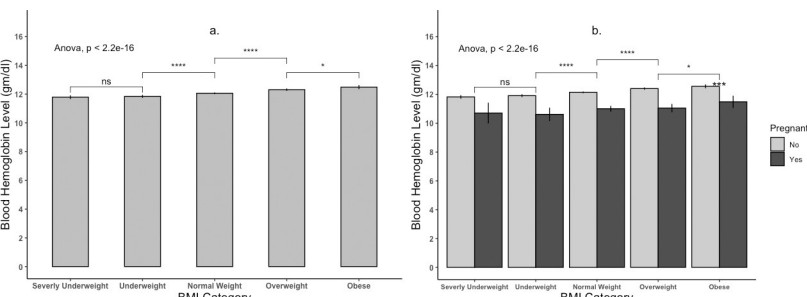

**Fig 1.** Hemoglobin level of Women (n = 5680) according to BMI Category (Picture a.) and Pregnancy Status (Picture b.).

**Table 4. Blood hemoglobin level of children (n = 2373) based on BMI SD score and gender category.**

| | Blood Hemoglobin Concentration (gm/dl) | | |
|---|---|---|---|
| | **Male†**<br>**[mean ± sd**<br>**(95%CI)]** | **Female†**<br>**[mean ± sd**<br>**(95%CI)]** | **Total**<br>**[mean ± sd**<br>**(95%CI)]** |
| *BMI Classification* | | | |
| *Severely Wasted (BMIZ<-3)* | 10.448±1.155(10.076–10.820)[a] | 10.643±1.417(10.135–11.150)[a] | 10.535±1.273(10.230–10.840)[a] |
| *Wasted (BMIZ<-2to≥-3)* | 10.666±1.206(10.446–10.886)[a] | 10.936±1.165(10.731–11.141)[a] | 10.805±1.190(10.655–10.956)[a] |
| *Normal Weight (BMIZ ≥ -2to≤ 2)* | 10.762±1.289(10.684–10.841)[a] | 10.838±1.195(10.763–10.912)[a] | 10.799±1.244(10.745–10.854)[a] |
| *Overweight (BMIZ >2to≤ 3)* | 11.220±1.367(10.372–12.067)[a] | 10.535±1.579(9.708–11.362)[a] | 10.820±1.503(10.219–11.422)[a] |
| *Obese (BMIZ>3)* | 11.618±1.513(10.723–12.512)[a] | 10.350±1.485(9.161–11.538)[a] | 11.170±1.584(10.417–11.924)[a] |
| **Anemia Classification** | | | |
| *Not anemic* | 11.788±0.638(11.736–11.840) | 11.757±0.644(11.705–11.810) | 11.773±0.641(11.736–11.810) |
| *Anemic* | 09.820±0.962(09.745–09.895) | 09.915±0.888(09.843–09.987) | 9.865±0.928(09.813–09.918) |
| **Anemia** | | | |
| *Not Anemic* | 11.788±0.638(11.736–11.840) | 11.757±0.644(11.705–11.810) | 11.773±0.641(11.736–11.810) |
| *Mild* | 10.497±0.283(10.467–10.527) | 10.487±0.290(10.457–10.518) | 10.429±0.286(10.471–10.514) |
| *Moderate* | 09.121±0.661(09.043–09.198) | 09.165±0.615(09.086–09.245) | 09.141±0.641(09.085–09.197) |
| *Severe* | 06.220±1.142(05.511–06.928) | 06.157±0.692(05.644–06.670) | 06.194±0.956(05.739–06.648) |
| **Total** | 10.755±1.283(10.683–10.827) | 10.837±1.204(10.768–10.906) | 10.795±1.245(10.745–10.845) |

N.B.: † indicate hemoglobin level between male and female children is not significantly different (p>0.05), Each column with different superscript indicates significant difference (p≤0.05) in hemoglobin level.

Predicted probability of being anemic from the binary (7a., 7b.) and ordinal (7c., 7d.) logistic regression are shown in Fig 7. The restricted cube splines (RCS) regression for women and children is shown in Fig 8. It is clear from the figure that the odds of being anemic were higher when the BMI was lower than the normal range for women. Conversely, as the BMI increases, the odds of being anemic were reported to be lower (Fig 8A). For Children, the findings of restricted cubic spline were inconsistent (Fig 8B).

## Discussion

Until recently, the evidence published in support of the association between BMI and anemia is controversial. Few studies reported direct association [21–27, 42], i.e., increasing BMI increases the chance of being anemic, while others reported the opposite [17, 18, 20], yet few

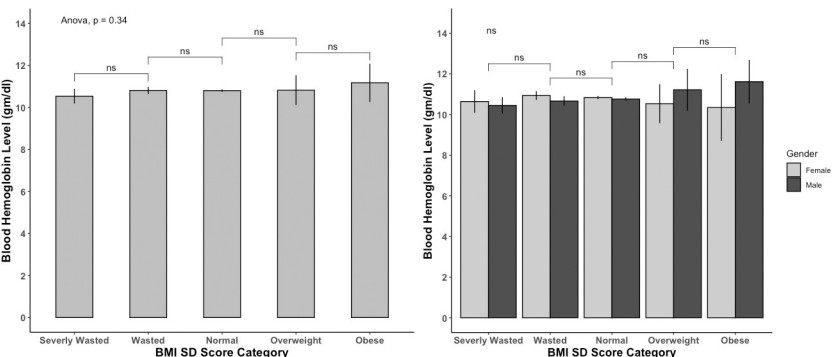

**Fig 2. Hemoglobin level of children (n = 2373) according to BMI category and gender.**

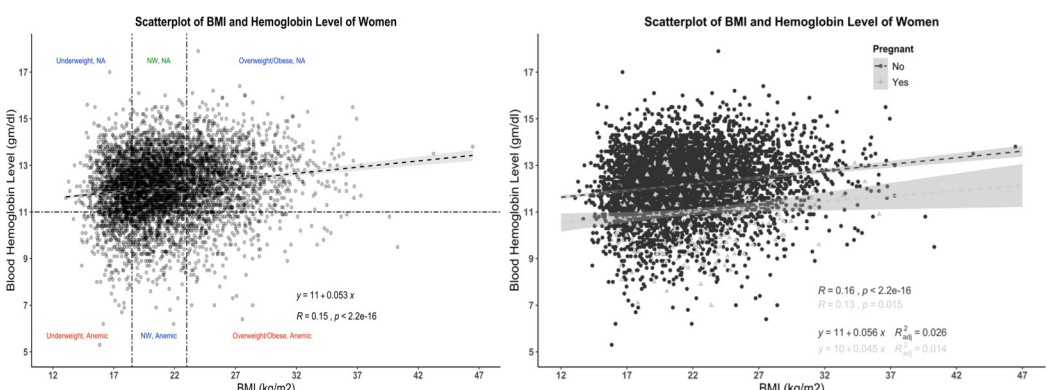

**Fig 3. Association (linear) of BMI and blood hemoglobin level of women (n = 5680).** Marginal density is shown for the three BMI categories (dotted vertical lines). Anemia is indicated using the cut-off <11 g/dl (dotted horizontal line).

others concluded no relationship between BMI and anemia [43]. However, there are discrepancies among studies in using anemia biomarkers. The current findings demonstrated that obese and overweight women have a lower probability of being anemic than normal and underweight women and these findings were consistent while using different statistical approaches.

Data from the national representative study was used in this study and the Two-way ANOVA analysis revealed that obese, and overweight women had significantly higher hemoglobin levels than normal and underweight women (Table 3). Likewise, when plotting BMI and blood hemoglobin level of women, a weak positive correlation (R = 0.15) was found, and this was similar for both pregnant (R = 0.13) and nonpregnant women (R = 0.16) (Fig 1). Unlike women, the plotting for children shown no correlation (Fig 2). A similar finding was reported by Ghose B et al. (2016), though they used only the Chi-square test and binary logistic regression without comparing blood hemoglobin level and focused on multiple factors of anemia [17]. Likewise, a cross-sectional study from China reported higher blood hemoglobin levels and a lower anemia prevalence ratio among obese and overweight women [18]. Another national representative study from Taiwan (NAHSIT) reported less than the half likelihood of being anemic among obese and overweight women than normal-weight women. Increased iron intake was reported among women with obesity, and overweight and from this finding, it may hypothesize that diet may link with increased hemoglobin levels among obese and overweight women [18]. However, in the current study, dietary iron intake is out of our focus;

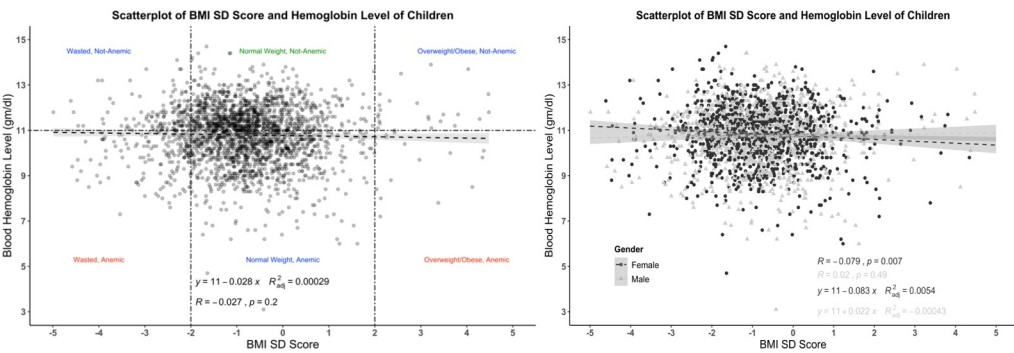

**Fig 4. Association (linear) of BMI and blood hemoglobin level of children (n = 2373).** Marginal density is shown for the three BMI categories (dotted vertical lines). Anemia is indicated using the cut-off <11 g/dl (dotted horizontal line).

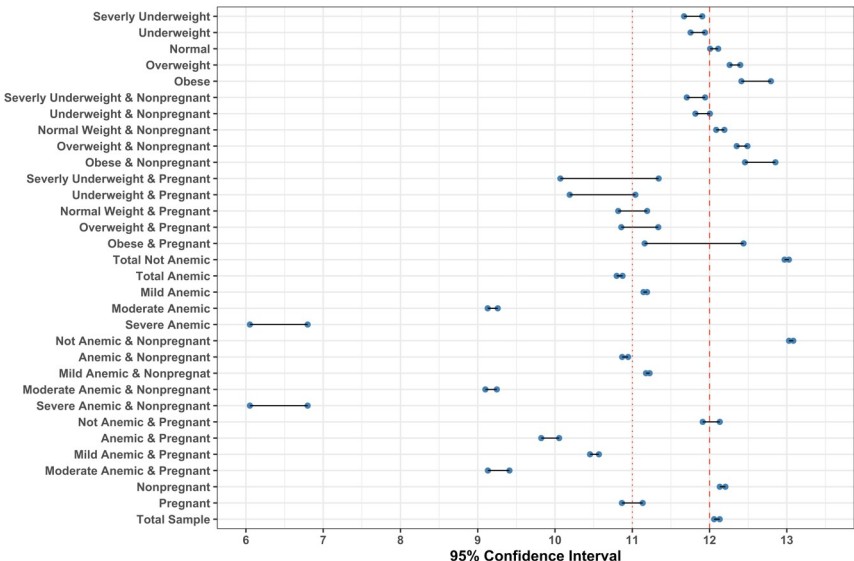

**Fig 5. Forest plot of 95% CI of blood hemoglobin level of women (n = 5680) of different BMI and anemia group (vertical lines are anemia cut-off point [red dotted line at 11 gm/dl]).**

thus, our findings cannot predict the relation between dietary iron intake and blood hemoglobin level.

On the contrary, when considering pregnancy status, only the findings from nonpregnant women were in the line of findings of total women. Hemoglobin level was not significantly different between pregnant obese/overweight women and normal/underweight women (Table 3 and Fig 5). This insignificant finding could be due to small percentages (~6%) of pregnant women than the nonpregnant women. Moreover, after categorizing women according to pregnancy status and BMI status, the number becomes further low in the obese (31 women),

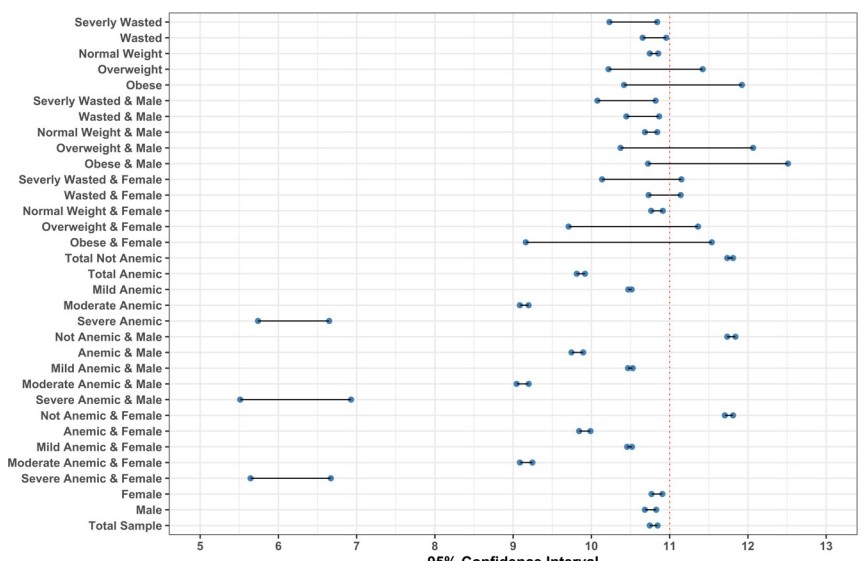

**Fig 6. Forest plot of 95% CI of blood hemoglobin level of children (n = 2373).** (Vertical lines are Anemia cut-off point [red dotted line at 11 gm/dl]).

Table 5. Results of multiple binary and ordinal logistic regression models of women (n = 5680) using anaemia status as binary and ordered response variable.

| | Binary Logistic Regression | | | | | | Ordinal Logistic Regression | | | | | | | |
| | (Not Anemic vs Anemic) | | | | | | (Not Anemic \| Mild \| Moderate \| Severe Anemic) | | | | | | | |
| | COR# (CI) | 95%CI of COR | p | AOR# (AIC#: 7552.8) | 95%CI of AOR | P | Wald (p) | Regression Coefficient | SE | p | AOR# (AIC#: 9719.9) | 95%CI of OR | Parallel Regression p | Omnibus p |
|---|---|---|---|---|---|---|---|---|---|---|---|---|---|---|
| Age (Years) | 1.00 | 1.000–1.011 | 0.0366 | 1.013 | 1.007–1.019 | 1.23e-05 | 19.1 | 0.0142 | 0.0029 | 1.491e-06 | 1.014 | 1.008–1.020 | 0.148 | 2.318e-14 |
| BMI (kg/m2) | 0.928 | 0.915–0.941 | 2e-16 | 0.938 | 0.923–0.953 | 2.80e-15 | (1.2e-05) | -0.06402 | 0.0079 | 5.994e-16 | 0.937 | 0.923–0.952 | 0.879 | (without Current Pregnancy 0.961194) |
| BMI (Category) | | | | | | | 61.4 | | | | | | | |
| Normal Weight (Ref) | 1 | | | 1 | | | | | | | | | | |
| Severely Underweight | 1.478 | 1.230–1.777 | 2.98e-05 | 1.401 | 1.163–1.589 | 0.000389 | (1.5e-12) | 0.3162 | 0.0911 | 5.182e-04 | 1.371 | 1.147–1.639 | 0.66 | |
| Underweight | 1.324 | 1.129–1.552 | 0.000538 | 1.279 | 1.088–1.502 | 0.002778 | | 0.2366 | 0.0794 | 2.916e-03 | 1.266 | 1.083–1.480 | 0.82 | |
| Overweight | 0.712 | 0.619–0.817 | 2.65e-08 | 0.732 | 0.638–0.794 | 8.16e-06 | | -0.3208 | 0.0688 | 3.145e-06 | 0.725 | 0.633–0.830 | 0.89 | |
| Obese | 0.553 | 0.445–0.685 | 3.17e-05 | 0.562 | 0.393–0.794 | 0.001344 | | -0.6016 | 0.1781 | 7.344e-04 | 0.547 | 0.383–0.771 | 0.85 | |
| Pregnant | | | | | | | 15.9 | | | | | | | |
| No (Ref) | 1 | | | 1 | | | | | | | | | | |
| Yes | 1.367 | 1.102–1.694 | 0.00429 | 1.565 | 1.250–1.958 | 6.54e-05 | (6.5e-05) | 0.7882 | 0.1156 | 9.416e-12 | 2.199 | 1.752–2.758 | 2.9292e-21 | |
| Wealth Index | | | | | | | 29.6 | | | | | | | |
| Middle(Ref) | 1 | | | 1 | | | | | | | | | | |
| Poorest | 1.281 | 1.078–1.523 | 0.00486 | 1.184 | 0.99–1.410 | 0.05606 | (5.9e-06) | 0.1626 | 0.9866 | 6.064e-02 | 1.176 | 0.992–1.394 | 0.895 | |
| Poorer | 1.192 | 1.005–1.414 | 0.04339 | 1.144 | 0.963–1.358 | 0.12489 | | 0.1419 | 0.0856 | 9.734e-02 | 1.152 | 0.974–1.363 | 0.968 | |
| Richer | 0.895 | 0.758–1.058 | 0.19611 | 0.953 | 0.805–1.128 | 0.58007 | | -0.5131 | 0.0842 | 5.423e-01 | 0.949 | 0.805–1.120 | 0.938 | |
| Richest | 0.626 | 0.529–0.739 | 3.63e-08 | 0.749 | 0.629–0.891 | 0.00155 | | -0.2741 | 0.0875 | 1.739e-03 | 0.766 | 0.640–0.902 | 0.655 | |
| Religion | | | | | | | 17.44 (0.00058) | | | | | | | |
| Islam(Ref) | 1 | | | 1 | | | | | | | | | | |
| Buddhism | 1.740 | 0.523–6.045 | 0.361 | 1.336 | 0.398–4.683 | 0.63534 | | 0.2660 | 0.5729 | 6.424e-01 | 1.304 | 0.406–3.987 | 0.994 | |
| Christianity | 0.828 | 0.330–1.938 | 0.672 | 0.841 | 0.332–1.989 | 0.70084 | | -0.2661 | 0.4380 | 5.434e-01 | 0.766 | 0.307–1.752 | 0.999 | |
| Hinduism | 1.422 | 1.204–1.680 | 3.41e-05 | 1.428 | 1.206–1.692 | 3.68e-05 | | 0.3549 | 0.0832 | 1.999e-05 | 1.426 | 1.210–1.677 | 0.968 | |
| Intercept$ (Not Anemic \| Mild Anemic) | | | | | | | | -0.5268 | 0.1867 | 4.780e-03 | | | | |
| Intercept$ (Mild Anemic \| Moderate Anemic) | | | | | | | | 1.8099 | 0.1909 | 2.561e-21 | | | | |
| Intercept$ (Moderate Anemic \| Severe Anemic) | | | | | | | | 5.7792 | 0.3985 | 1.228e-47 | | | | |

# COR = Crude Odd Ratio, AOR = Adjusted Odd Ratio, AIC = Akaike Information Criterion;

$ Intercept from the Ordinal Logistic Regression.

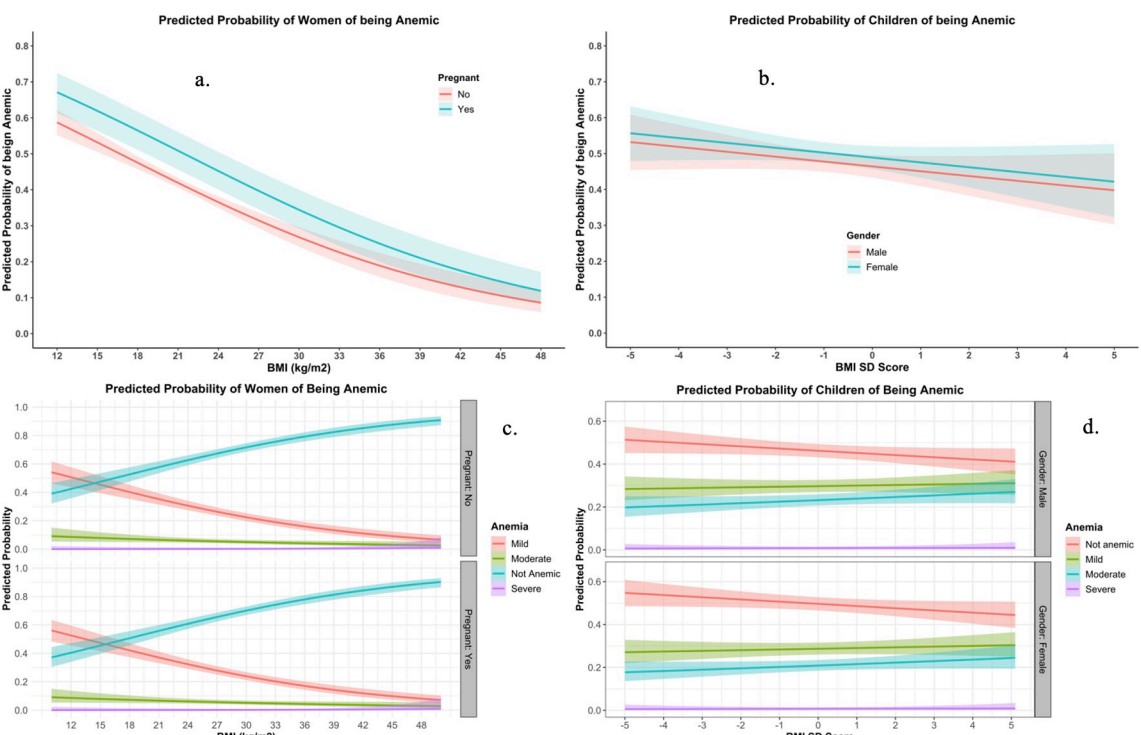

**Fig 7.** Predicted Probability of Anemia of women (Fig a., c.) and Children (Fig b., d.) from Binary Logistic (Fig a. b.) and Ordinal Logistic regression (Fig c., d.). All of the models were adjusted for Age, Pregnancy, Gender and Wealth Index.

underweight (39 women), and severely underweight (18 women) group. Furthermore, studies are limited to demonstrate the impact of BMI categories on anemia and hemoglobin level of pregnant women. To the best of our knowledge, unlike our findings, Mocking M et al. (2018) reported significantly higher hemoglobin levels and a lower risk of anemia among early pregnant women with higher BMI [44]. Thus, this small number of samples in each group may not be representative. Further study with a larger sample may be required to elucidate the actual difference in hemoglobin level among different BMI categories of pregnant women.

Binary and ordinal logistic regression from the current study shows that obese ad overweight women have a lower risk of anemia compared to normal and underweight women (Table 5). Few covariates like age, BMI, pregnancy status, wealth index, and religion were

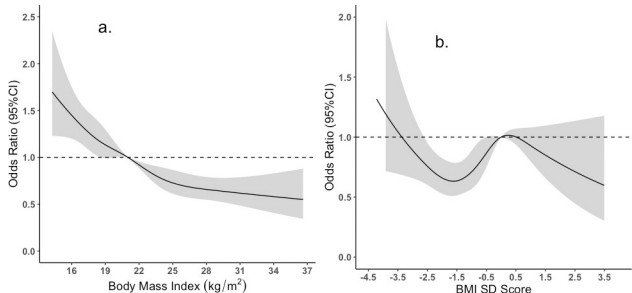

**Fig 8.** The Odd of being anemic from Restricted Cubic Splines (RCS) regression for Women (Picture a.) and Children (Picture b.). [The reference BMI for women was set at 21 (median of BMI for normal weight) and for Children reference BMI SD score at 0].

**Table 6. Results of multiple binary and ordinal (proportional odd model) logistic regression models of children (n = 2373) using anaemia status as binary and ordered response variable.**

| | Binary Logistic Regression (Not Anemic vs Anemic) | | | | | | | Ordinal Logistic Regression (Not Anemic \| Mild \| Moderate \| Severe Anemic) | | | | | | |
|---|---|---|---|---|---|---|---|---|---|---|---|---|---|---|
| | COR[#] (CI) | 95%CI of COR | p | AOR[#] (AIC[#]: 3065.9) | 95%CI of AOR | P | Wald (p) | Regression Coefficient | SE | p | AOR[#] (AIC[#]: 4827.8) | 95%CI of OR | Parallel Regression p | Omnibus p |
| Age (Months) | 0.9633 | 0.958–0.9688 | 2e-16 | 0.9624 | 0.957–0.967 | 2e-16 | 179.9 (0.000) | -0.0403 | 0.0026 | 1.655e-52 | 0.9604 | 0.955–0.965 | 0.2023 | 0.4837 |
| **Gender** | | | | | | | | | | | | | | |
| Male (Ref) | 1 | | | 1 | | | | | | | | | | |
| Female | 0.9038 | 0.769–1.061 | 0.2184 | 0.8804 | 0.742–1.043 | 0.1411 | 2.2 (0.14) | -0.1554 | 0.0795 | 0.05063 | 0.8560 | 0.732–1.000 | 0.4874 | |
| BMI SD Score | 1.0565 | 0.985–1.132 | 0.1198 | 1.0436 | 0.969–1.123 | 0.2570 | 1.3 (0.26) | 0.0253 | 0.0344 | 0.4618 | 1.0256 | 0.958–1.097 | 0.0846 | |
| **BMI (Category)** | | | | | | | | | | | | | | |
| Normal Weight (Ref) | 1 | | | 1 | | | 4.2 | | | | | | | |
| Severely Wasted | 1.4964 | 0.914–2.492 | 0.1134 | 1.2680 | 0.755–2.161 | 0.3741 | (0.38) | 0.2906 | 0.2341 | 0.2145 | 1.3372 | 0.842–2.115 | 0.8965 | |
| Wasted | 0.9489 | 0.725–1.240 | 0.7014 | 0.9654 | 0.727–1.281 | 0.8078 | | -0.0365 | 0.1332 | 0.7840 | 0.9641 | 0.741–1.250 | 0.9191 | |
| Overweight | 0.8029 | 0.351–1.803 | 0.5944 | 0.6816 | 0.287–1.592 | 0.3746 | | -0.3620 | 0.4052 | 0.3716 | 0.6962 | 0.305–1.517 | 0.9440 | |
| Obese | 0.3954 | 0.125–1.070 | 0.0824 | 0.4038 | 0.120–1.177 | 0.1119 | | -0.6842 | 0.5552 | 0.2177 | 0.5044 | 0.153–1.411 | 0.4647 | |
| Wealth Index | | | | | | | | | | | | | | |
| Middle (Ref) | 1 | | | 1 | | | | | | | | | | |
| Poorer | 1.2365 | 0.950–1.609 | 0.1141 | 1.8753 | 0.901–1.565 | 0.2217 | 44.4 | 0.2582 | 0.1280 | 0.043689 | 1.2946 | 1.007–1.664 | 0.3386 | |
| Poorest | 1.3271 | 1.029–1.711 | 0.0291 | 1.3838 | 1.061–1.805 | 0.0165 | (5.2e-09) | 0.3399 | 0.1224 | 0.005516 | 1.4048 | 1.105–1.787 | 0.7765 | |
| Richer | 0.7486 | 0.573–0.975 | 0.0326 | 0.7198 | 0.544–0.950 | 0.0205 | | -0.2712 | 0.1316 | 0.039438 | 0.7624 | 0.588–0.986 | 0.7711 | |
| Richest | 0.7364 | 0.567–0.955 | 0.0215 | 0.6549 | 0.497–0.861 | 0.0024 | | -0.3544 | 0.1300 | 0.006432 | 0.7015 | 0.543–0.905 | 0.2484 | |
| **Intercept**[$] (Not Anemic \| Mild Anemic) | | | | | | | | -1.4972 | 0.1370 | 7.445e-20 | | | | |
| **Intercept**[$] (Mild Anemic \| Moderate Anemic) | | | | | | | | -0.0563 | 0.1333 | 0.78871 | | | | |
| **Intercept**[$] (Moderate Anemic \| Severe Anemic) | | | | | | | | 3.7365 | 0.2706 | 1.082e-39 | | | | |

[#]COR = Crude Odd Ratio, AOR = Adjusted Odd Ratio, AIC = Akaike Information Criterion; [$]Intercept from the Ordinal Logistic Regression.

included in the logistic regression model as these factors are supposed to be linked with BMI. Both adjusted and crude odd ratios provided similar results. As age increases, the probability of being anemic is shown to be higher for women (OR 1.006; 95%CI 1.000–1.011) (Table 5). However, the opposite was found for children (OR 0.96, 95%CI 0.958–0.9688) (Table 6). Nevertheless, it is predicted that as age increases, BMI increases. Thus, older women would have higher BMI, and the risk of anemia would be lower. However, this finding is counterintuitive to our findings. For example, Jackson et al. (2004) reported that older women greater than 50 years of age had higher blood hemoglobin levels and a lower risk of anemia than women lower than 50 years [45].

From the binary and ordinal logistic regression, it is clear that the risk of being anemic is lower among women with the highest wealth quintile (OR 0.626, 95% CI 0.529–0.739) compared to women with lower (OR 1.192, 95% CI 0.529–0.739) and middle wealth quintile

(Table 5). Thus, BMI is either directly or indirectly linked with the wealth index, and the prevalence of obesity is higher among people of the higher wealth quintile [46]. Therefore, when shedding light on anemia, BMI, and wealth quintile, the relation of lower anemia risk among obese women and lower anemia risk among richest women can be easily linked. During the last few decades, Bangladesh faces demographic, economic, and nutrition transition, which has led to changes in dietary habits, particularly among the higher socioeconomic groups [47, 48]. Along with the higher prevalence of obesity among higher socioeconomic groups, underweight/undernutrition is still higher predominantly among lower socioeconomic groups [49, 50]. Therefore, it is very likely that dietary iron intake, exclusively the bioavailable heme iron from food of animal origin, is higher among obese and overweight women than underweight women. This higher iron intake eventually leads to higher blood hemoglobin levels and lower anemia risk among obese and underweight women. Higher iron intake among obese women has also been reported by Yu Qin et al. (2013) among Chinese women compared to underweight women [18].

Nevertheless, several studies reported a higher probability of anemia among obese women, though they reported higher blood hemoglobin levels and lower blood iron and ferritin levels [20, 34, 51, 52]. The higher prevalence of anemia among obese women is sometimes explained as an effect of either dilutional hypoferremia, poor dietary iron intake, increased iron requirements, and/or impaired iron absorption in obese individuals [31] and higher inflammation due to higher hepcidin levels, which reduces iron absorption from the intestine, is supposed to be most plausible [53]. However, in most studies, higher blood hemoglobin levels with lower blood iron remain inconclusive and need further studies. Furthermore, obesity and overweight are sometimes linked with intake of energy-dense food with limited micronutrients, referred to as hidden hunger [54]. Few studies reported a similar levels of dietary iron intake among obese/overweight adults [55] and children [56]. One recent study reported that overweight and obese women who consume higher fat than carbohydrates are prone to develop iron deficiency anemia than those who consume more carbohydrates [20]. From this finding, we may hypothesize that Bangladesh is in the early stage of nutrition transition [57], and the ratio of dietary fat/carbohydrate is not high enough to lessen the protective effect of obesity on anemia.

When the predicted probability of anemia was plotted against BMI using binary logistic regression (Fig 7A), it is clear that the probability becomes lowered as the BMI increases. Nonetheless, when using ordinal logistic regression to plot the probability of different stages of anemia, a diverse picture of probability was found (Fig 7C). The probability of severe and moderate anemia was shown to be identical over the range of BMI, but only the probability of mild anemia becomes lowered as BMI increases, and the probability of being anemic becomes lowered. These discrepant findings could be explained that low-grade inflammation among moderate and severe anemic individuals becomes dominant over higher dietary intake of iron among obese/overweight women. Moreover, the prevalence of obesity is not as high in Bangladesh right now to shift the inflammatory response mechanism dominant over higher dietary intake as morbid obesity has been identified as a frequent predictor of iron deficiency [58].

The strength of the current study is that it includes a sizeable nationwide representative sample from Bangladesh. However, there are few limitations of our study, which need to be considered during future studies. First of all, hemoglobin, the only hematological biomarker marker, was taken without considering other biomarkers like serum ferritin, transferrin, iron, and inflammatory biomarkers like CRP, hepcidin, and others. Therefore, it is difficult to explain the inconsistencies in different study findings regarding anemia prevalence, hemoglobin level, and iron level among obese and underweight individuals with a single anemia biomarker. Secondly, the number of obese/overweight and underweight women among different

anemia categories was lower, which might be a problem in concluding the probability of different categories of anemia among women with different BMI categories. Thirdly, the data used in this study is secondary, which means we had no control over the study design and measurement procedure.

Moreover, the national representative data dates back from 2011, which means the pattern of anemia prevalence might have been changed over the period. Therefore, the current study urges the necessity to include multiple anemia biomarkers in future studies to address those limitations and better explain the relationship. However, the current study might help policy-makers address anemia as a public health problem in Bangladesh.

## Conclusion

In conclusion, our study on Bangladeshi women established that as the BMI increases, blood hemoglobin level increases, and the risk of anemia decreases. Among children, no association between BMI and anemia was not found. Like women with higher BMI, anemia's prevalence was lowered among the richest women, indicating higher dietary iron intake among the richest women, and richness is associated with higher BMI. This study contributed to the existing knowledge regarding the association between BMI and anemia risk. However, the current study urges the necessity of further study to include multiple hematological biomarkers and inflammatory biomarkers to elucidate better the complex association between BMI and anemia, which is still contentious.

## Author Contributions

**Conceptualization:** Md Kamruzzaman.

**Data curation:** Md Kamruzzaman.

**Formal analysis:** Md Kamruzzaman.

**Software:** Md Kamruzzaman.

**Validation:** Md Kamruzzaman.

**Visualization:** Md Kamruzzaman.

**Writing – original draft:** Md Kamruzzaman.

**Writing – review & editing:** Md Kamruzzaman.

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
