## [Decision Letter · Decision Letter 0]

22 Jul 2021

PONE-D-21-16964

Is BMI associated with anemia and hemoglobin level of women and children in Bangladesh: A study with multiple statistical approach

PLOS ONE

Dear Dr. Kamruzzaman,

Thank you for submitting your manuscript to PLOS ONE. After careful consideration, we feel that it has merit but does not fully meet PLOS ONE’s publication criteria as it currently stands. Therefore, we invite you to submit a revised version of the manuscript that addresses the points raised during the review process.

The paper has been reviewed by three independent reviewers. They have raised some concerns those need to be fixed before taking final decision. 

We look forward to receiving your revised manuscript.

Kind regards,

Enamul Kabir

Academic Editor

PLOS ONE

Journal Requirements:

Reviewers' comments:

Reviewer's Responses to Questions

**Comments to the Author**

1. Is the manuscript technically sound, and do the data support the conclusions?

Reviewer #1: Yes

Reviewer #2: Yes

Reviewer #3: Yes

2. Has the statistical analysis been performed appropriately and rigorously? 

Reviewer #1: No

Reviewer #2: Yes

Reviewer #3: Yes

3. Have the authors made all data underlying the findings in their manuscript fully available?

Reviewer #1: Yes

Reviewer #2: No

Reviewer #3: No

4. Is the manuscript presented in an intelligible fashion and written in standard English?

Reviewer #1: Yes

Reviewer #2: Yes

Reviewer #3: Yes

5. Review Comments to the Author

Reviewer #1: The manuscript was brilliantly written, the objectives of the research were met. Manuscript has intelligently communicated to the readers But I would appreciate if the authors give more clarification to the statistical test used for analyzing the correlation between haemoglobin level of the women (according to their pregnancy status) and their BMI categories. I would suggest that Two-way ANOVA test is more appropriate than One-Way ANOVA because you are dealing with two dependent variables that have more three categories.

Secondly, in discussion column, the correlation between risk of being anemic and wealth. I need the authors to clarify what factor was used to determine the level of wealth among the research participants, is the amount of income earned or their lifestyles? I believe the amount of income earned is a more appropriate factor compared to the level of wealth.If these concerns raised are addressed, I believe the manuscript is good for acceptance by the editorial board. Once again, the authors did brilliantly

Reviewer #2: The study is an important addition to the already existing knowledge on BMI, anemia and hemoglobin level in women and children as it aimed to establish the association between BMI, and anemia and hemoglobin among women and children. However, the comments below should be addressed to enhance the quality of the paper.

Abstract:

The main objective of the study is not stated in the background of the abstract. The main objective of the study should be clearly stated in the abstract.

The paper also fails to clearly state the methods which were adopted in undertaking the study. The sample size of the study, the multiple statistical approaches, etc. which were used in the study is not stated in the methods of the abstract.

General comments:

There are some grammatical errors in the manuscript including missing commas, missing function words and dots. E.g. (lines; 48, 81, 87, 107, 116, 117, 285, 314, 316, 320, 324, 327, 392-393, 395, etc.). Author should carefully read through the paper and correct all grammatical errors.

In the introduction and throughout the paper, some sentences are not well structured and therefore makes them difficult to understand. Missing and misplaced commas, and function words makes some sentences ambiguous. Long and complex sentences have also been used in the manuscript E.g. (lines; 98-101, 101-102, 105-107, 118-121 etc.) Author should carefully read through the paper and restructure long and complex sentences to make them simple and easy to comprehend by a lay person.

The objective of the study was not clearly stated in the background of the paper.

The materials and methods used for the study is carefully written by the author. The kind of data used as well as the size of data on women and children from the secondary data which was analyzed by the author was clearly stated. The different forms of statistical analyses which were performed in the study were described in the methods. However, the eligibility criteria which was utilize in the selection of study participants was not clearly described in the methods. Author should be clear on the eligibility criteria utilized in the study.

The author has carefully presented the results and discussion of the study. The findings have been compared and contrasted with other existing findings. However, the discussion of this paper fails to capture the policy influence as well as the influence of the study findings on future studies relating to BMI, anemia and hemoglobin levels of women and children.

Reviewer #3: The author should address the grammeatical and typographical errors in the abstract and other parts of the manuscript.

Is the study a restrospective or a prospective study

If prospective then there is no need to highlight how the haemoglobin was estimated in detail

The authors did not also establish whether other causes of factors responsible for anaemia were excluded

6. PLOS authors have the option to publish the peer review history of their article (what does this mean?). If published, this will include your full peer review and any attached files.

Reviewer #1: **Yes: **Emmanuel Adamolekun

Reviewer #2: No

Reviewer #3: No

---

## [Author Response · Author response to Decision Letter 0]

14 Aug 2021

REBUTTAL LETTER 

14 August 2021

Enamul Kabir

Academic Editor

PLOS ONE

Ref: Manuscript ID: PONE-D-21-16964

Manuscript Title: Is BMI associated with anemia and hemoglobin level of women and children in Bangladesh: a study with multiple statistical approaches

Dear Enamul Kabir, 

Thank you for your email and for considering our submission. We have addressed the issues raised by the editorial team. A rebuttal letter, which provides our responses, is appended. We hope that the revised manuscript will prove acceptable for publication.

Kind Regards,

Md Kamruzzaman

POINT-BY-POINT REBUTTAL

Response to Editor

Response: We thank the Associate Editor for this comment. We have checked again the PLOS ONE’S style requirements and ensured the style.

Response: We thank the editor for this comment. The data is secondary and available for use by all. However, there are ethical restriction on sharing the dataset. Anyone wants to use the dataset can register and use. 

Details of information, that are requested to access the dataset, are given below.

The dataset is available on URLs:

https://www.dhsprogram.com/data/available-datasets.cfm

General Instructions: Exploring the above webpage, individuals first need to register as a new user to access the dataset. After registration, new project should be created and there the country list should be provided. On the following webpages, following step-by-step the user should be registered first, and then followed the instruction on screen after sign-in to access the dataset.

Contact Person and Email Address: 

Name: Gbaike Ajay

Email Address: Gbaike.Ajayi@icf.com

Data Access

For questions or comments about accessing 

The DHS Program data, please contact 

Email: archive@dhsprogram.com.

Comments from Reviewer 1 

Reviewer #1: 

Q1: The manuscript was brilliantly written; the objectives of the research were met. Manuscript has intelligently communicated to the readers But I would appreciate if the authors give more clarification to the statistical test used for analyzing the correlation between haemoglobin level of the women (according to their pregnancy status) and their BMI categories. I would suggest that Two-way ANOVA test is more appropriate than One-Way ANOVA because you are dealing with two dependent variables that have more three categories.

Response: We thank the reviewer for their comments. We have revised the discussion section and provided more information (Page 7, Line 357). 

Q2: Secondly, in discussion column, the correlation between risk of being anemic and wealth. I need the authors to clarify what factor was used to determine the level of wealth among the research participants, is the amount of income earned or their lifestyles? I believe the amount of income earned is a more appropriate factor compared to the level of wealth.If these concerns raised are addressed, I believe the manuscript is good for acceptance by the editorial board. Once again, the authors did brilliantly

Response: We thank the reviewer for their comments. Wealth index has been developed considering household asset and household income using principledly component analysis. Direct household income data is not available. Moreover, household wealth indirectly indicates household income and sometimes better explain household financial status, rather considering only disposable income. We have revised the discussion section and provided more information (Page 6, Line 343). 

Comments from Reviewer 2 

Reviewer #2: 

Q1: Abstract:

The main objective of the study is not stated in the background of the abstract. The main objective of the study should be clearly stated in the abstract.

Response: Thanks for this comment. The abstract section has been revised accordingly (Page 2, Line 51).

Q2: The paper also fails to clearly state the methods which were adopted in undertaking the study. The sample size of the study, the multiple statistical approaches, etc. which were used in the study is not stated in the methods of the abstract.

Response: Thanks for this comment. The abstract section has been revised accordingly (Page 2, Line 55).

Q3: General comments:

There are some grammatical errors in the manuscript including missing commas, missing function words and dots. E.g. (lines; 48, 81, 87, 107, 116, 117, 285, 314, 316, 320, 324, 327, 392-393, 395, etc.). Author should carefully read through the paper and correct all grammatical errors.

Response: Thanks for this comment. The manuscript has been revised and corrected accordingly.

Q4: In the introduction and throughout the paper, some sentences are not well structured and therefore makes them difficult to understand. Missing and misplaced commas, and function words makes some sentences ambiguous. Long and complex sentences have also been used in the manuscript E.g. (lines; 98-101, 101-102, 105-107, 118-121 etc.) Author should carefully read through the paper and restructure long and complex sentences to make them simple and easy to comprehend by a lay person.

Response: Thanks for this comment. The manuscript has been revised and corrected accordingly.

Q5: The objective of the study was not clearly stated in the background of the paper.

Response: The introduction section has been revised accordingly (Page 5, Line 244).

Q2: Please add in the previous findings on associations between anthropometric and nutritional status, identify the gaps in current literature, and what is the significance of this study.

Response: Thanks for this comment. This issue has been addressed (Page 4, Line 135).

Q4 Materials and Methods:

The materials and methods used for the study is carefully written by the author. The kind of data used as well as the size of data on women and children from the secondary data which was analyzed by the author was clearly stated. The different forms of statistical analyses which were performed in the study were described in the methods. However, the eligibility criteria which was utilize in the selection of study participants was not clearly described in the methods. Author should be clear on the eligibility criteria utilized in the study.

Response: Thanks for this comment. Ethical approval no has been added to the method section (Page 5, Line 259). 

Q5: The author has carefully presented the results and discussion of the study. The findings have been compared and contrasted with other existing findings. However, the discussion of this paper fails to capture the policy influence as well as the influence of the study findings on future studies relating to BMI, anemia and hemoglobin levels of women and children.

Response: Thanks for raising this nice issue. Indeed, it is valuable to include policy implication and future prospect of this study. These issues have been added in the discussion section. (Page 13, Line 771)

Comments from Reviewer 2 

Q1: The author should address the grammeatical and typographical errors in the abstract and other parts of the manuscript.

Response: Thanks for this comment. The manuscript has been revised and corrected accordingly. 

Q2: Is the study a retrospective or a prospective study, if prospective then there is no need to highlight how the haemoglobin was estimated in detail.

Response: Thanks for this comment. First of all, this is not a cohort study, rather cross-sectional. Thus, it’s not necessary to classify as prospective or retrospective study. Details of hemoglobin measurement has been highlighted to give an idea to the readers. 

Q3: The authors did not also establish whether other causes of factors responsible for anaemia were excluded

Response: Thanks for this comment. It’s true that, we have not addressed other factors that may cause anemia. We had only focused our light on the relationship between BMI and anemia and hemoglobin level. Thus, without including other factors in the current study, it’s not necessary to establish other factors. However, other common causes of anemia are already known. We have addressed the necessity of including other inflammatory factor in the future studies, those still need to be elucidated.

---

## [Decision Letter · Decision Letter 1]

13 Oct 2021

Is BMI associated with anemia and hemoglobin level of women and children in Bangladesh: a study with multiple statistical approaches

PONE-D-21-16964R1

Dear Dr. Kamruzzaman,

We’re pleased to inform you that your manuscript has been judged scientifically suitable for publication and will be formally accepted for publication once it meets all outstanding technical requirements.

Kind regards,

Enamul Kabir

Academic Editor

PLOS ONE

Additional Editor Comments (optional):

Reviewers' comments:

Reviewer's Responses to Questions

**Comments to the Author**

1. If the authors have adequately addressed your comments raised in a previous round of review and you feel that this manuscript is now acceptable for publication, you may indicate that here to bypass the “Comments to the Author” section, enter your conflict of interest statement in the “Confidential to Editor” section, and submit your "Accept" recommendation.

Reviewer #1: All comments have been addressed

Reviewer #2: All comments have been addressed

2. Is the manuscript technically sound, and do the data support the conclusions?

Reviewer #1: Yes

Reviewer #2: Yes

3. Has the statistical analysis been performed appropriately and rigorously? 

Reviewer #1: Yes

Reviewer #2: Yes

4. Have the authors made all data underlying the findings in their manuscript fully available?

Reviewer #1: Yes

Reviewer #2: Yes

5. Is the manuscript presented in an intelligible fashion and written in standard English?

Reviewer #1: Yes

Reviewer #2: Yes

6. Review Comments to the Author

Reviewer #1: I appreciate the facts the authors took to correction the issues raised in the first review and explained more details about the statistical tests used and grammatical errors. I believe this manuscript can be published

Reviewer #2: The author has carefully addressed most of the issues raised. However, there are some few typographical errors and needs to be corrected before the publication of the manuscript.

7. PLOS authors have the option to publish the peer review history of their article (what does this mean?). If published, this will include your full peer review and any attached files.

Reviewer #1: No

Reviewer #2: No

---

## [Editor Report · Acceptance letter]

19 Oct 2021

PONE-D-21-16964R1 

Is BMI associated with anemia and hemoglobin level of women and children in Bangladesh: a study with multiple statistical approaches 

Dear Dr. Kamruzzaman:

I'm pleased to inform you that your manuscript has been deemed suitable for publication in PLOS ONE. Congratulations! Your manuscript is now with our production department. 

Kind regards, 

on behalf of

Dr. Enamul Kabir 

Academic Editor

PLOS ONE